# Quest2ROS: An App to Facilitate Teleoperating Robots

**Nils Ingelhag***
ingelhag@kth.se
KTH Royal Institute of Technology
Stockholm, Sweden

**Michael C. Welle***
mwelle@kth.se
KTH Royal Institute of Technology
Stockholm, Sweden

**Martina Lippi**
martina.lippi@uniroma3.it
Roma Tre University
Rome, Italy

**Maciej Wozniak**
maciejw@kth.se
KTH Royal Institute of Technology
Stockholm, Sweden

**Andrea Gasparri**
andrea.gasparri@uniroma3.it
Roma Tre University
Rome, Italy

**Danica Kragic**
dani@kth.se
KTH Royal Institute of Technology
Stockholm, Sweden

## ABSTRACT

Teleoperation is an integral part of robotics research. In this work, we present Quest2ROS, a stand-alone app available for Oculus Quest 2 and 3 that facilitates the teleoperation of robots via ROS. Quest2ROS publishes the position and velocity of both hand-held controllers, the button pressed as well as enables haptic feedback via the controller vibration. The Quest headset does not have to be worn at teleoperation time to not restrict the operator, furthermore, a simple way to align the coordinate frame of the controller with any given robot in the real world is provided. We measure the tracking accuracy of a Quest 2 to be 0.46 mm on average, with mean latency between the Quest 2 and a ROS node being 82 ms and update frequency of relevant ROS topics being 71.96 Hz.

**ACM Reference Format:**
Nils Ingelhag*, Michael C. Welle*, Martina Lippi, Maciej Wozniak, Andrea Gasparri, and Danica Kragic. 2024. Quest2ROS: An App to Facilitate Teleoperating Robots. In *7th International Workshop on Virtual, Augmented, and Mixed-Reality for Human-Robot Interactions at HRI 2024.* ACM, Boulder, CO, USA, 5 pages.

## 1 TELEOPERATION OF ROBOTS

Teleoperation is fundamental in multiple robotics applications. To mention a few, it can be used to remotely operate robots in hazardous environments [1, 2], to perform demonstrations of wanted trajectories to bootstrap different learning approaches such as DAGGER [3] or Diffusion Policy [4]. A successful teleoperation system should mainly be *i)* intuitive, *ii)* robot agnostic, and *iii)* cost-efficent. To fully teleoperate a manipulator, its Cartesian coordinates, i.e., position and orientation of the end effector, should be controlled. However, operating intuitively in the Cartesian space is rather challenging. One common, and cost-efficient solution is the use of a computer mouse adapted to work in 3D space as demonstrated in [5]. While experienced users can perform impressive teleoperation tasks with such a setup [4], the user does not directly mimic the

---

*These authors contributed equally (listed in alphabetical order).

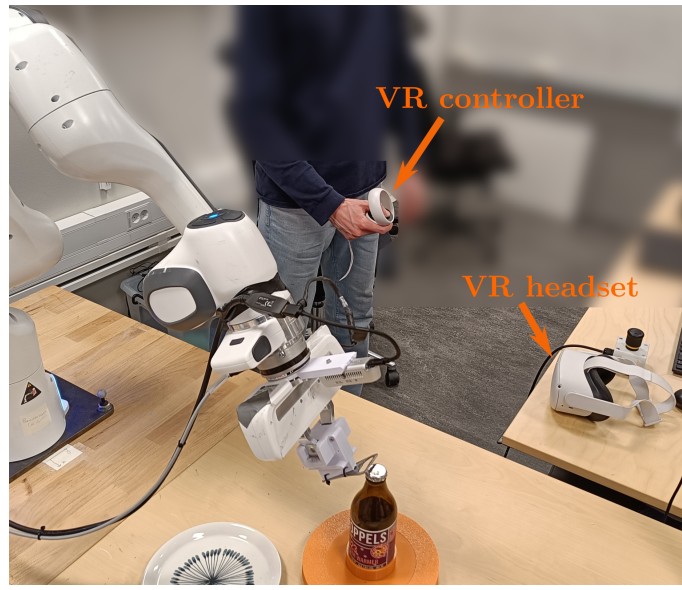

**Figure 1: Example usage of the Quest2ROS teleoperation on a Franka Panda robot. The headset is placed on the table with an unobstructed view of the handheld controller, which in turn is used to teleoperate the robot.**

movement of the robot in the Cartesian space. Other works address this by following a master-slave or *puppeteer* setup where the robotic system in question consists of a master robot that can be manually moved and the movements are mirrored to the slave robot or *puppet*. While this approach excels in achieving precise teleoperation and has proven its effectiveness in handling delicate tasks such as lid opening, juggling and cable tie zipping in [6], its implementation is generally costly. Other teleoperation systems are highly specific and only applicable for the robot they were developed for such as [7–9].

In recent years, numerous Augmented Reality (AR) or Virtual Reality (VR) teleoperation projects for human-robot interaction have emerged, due to advancements in communication technologies and Virtual and Augmented Reality (VAM) hardware [10]. These projects have been developed for a variety of tasks, including cloth folding [11], pick-and-place operations [12], and diagnostics and debugging [13]. Additionally, VAM technologies have been used for demonstration learning [14] or enhancing action explainability and

**Figure 2: Overview of Quest2ROS architecture. The VR controller (a) is tracked by the VR headset (b). The telemetry is streamed by the headset via Wi-Fi to a TCP endpoint (c) running on a specified host. The data is then made available in ROS via different topics (d). The telemetry data can then be used to control a robot (e) via ROS subscriber nodes.**

human-robot understanding [15, 16]. Additionally, many studies such as [17] and [18] have made significant efforts to enhance the visualization of robot sensory information within AR or VR environments. While these projects serve as impressive proof of concepts, their developers often had to begin from scratch due to the absence of required software and specific tools. Moreover, these projects tend to be specific to particular VAM and robot hardware.

In this work, we present Quest2ROS, an app that facilitates intuitive robot-agnostic teleoperation on consumer-grade hardware using ROS middleware. In detail, the app is available for the Oculus Quest 2 and 3 and allows users to use the VR controller to teleoperate the robot, as shown in Fig. 1. The VR headset needs to be placed with an unobstructed view of the VR controller for optimal tracking performances. The app is available in Metas Applab environment via an alpha version invitation link[1]. Further instructions, demonstration/evaluation videos as well as links to the ROS code repository can be found on the project's website[2].

## 2   QUEST2ROS

Quest2ROS is a stand-alone app for Oculus 2 and 3 that lets the user easily retrieve the position, velocity, button inputs as well as controlling the haptic feedback of both hand controllers via ROS. It is based on the teleoperation framework presented in [19]. The Oculus app streams telemetry data via Wi-Fi to a TCP-endpoint running on a host specified by the user. The streamed controller data is then made available via ROS topics to other ROS nodes. An overview of the system is shown Figure 2. To make the app more usable for controlling physical robots and actuators, the app is provided with built-in functionality to let the user set the reference frame in which the VR controller's pose and twist information are given.

The app is implemented using the Unity game engine together with the Oculus Integration[3] and ROS-TCP-endpoint[4] packages. The position and velocity of the controllers are continuously tracked by the headset using a combination of internal sensors and visual tracking. It is therefore important that there exists no visual occlusion between the headset and the controllers at all times during operation.

---

[1]https://www.meta.com/s/1stEAnW1u
[2]https://quest2ros.github.io/q2r-web/
[3]https://developer.oculus.com/downloads/package/unity-integration/
[4]https://github.com/Unity-Technologies/ROS-TCP-Connector

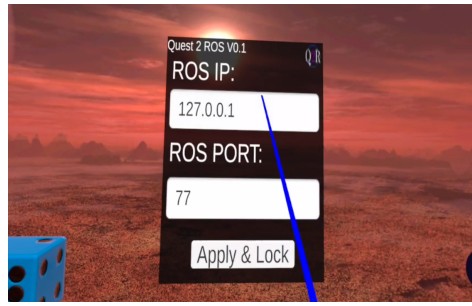

**Figure 3: The figure shows the configuration window within the app. The window lets the user set the IP address and port of the TCP endpoint.**

### 2.1   User interface (UI)

The user interface is a configuration window accessible within the VR world of the app. It lets the user set the IP and port of the TCP to ROS gateway running on a host. This configuration has to match the configuration to the TCP-to-ROS endpoint. The configuration is by default locked to prevent accidental reconfiguration but can be unlocked by pressing A and B buttons, and both triggers on the right VR controller. The IP and port can then be set using a virtual keyboard. After both the IP and port are set, the user has to press Apply & Lock to set the configuration as well as to lock the screen again. An image of the user interface is shown in Figure 3.

### 2.2   Coordinate frame alignment

The coordinate frame alignment lets the user set a new reference frame based on the position of one of the controllers. All pose and twist information is thereafter given in reference to the newly set coordinate system. The purpose of the coordinate frame alignment functionality is to make it easy for users to use the controller telemetry for robotic teleportation without complex coordinate transformations. Since the initial coordinate system is arbitrarily set to the momentary position of the VR headset at startup, coordinate frame alignment lets the user align the coordinate system to a physical stationary system within the room, e.g., with the robot coordinate system. The process of coordinate system alignment is shown in Figure 4. Either controller is aligned to the desired reference frame and the buttons A and B for the right controller or X and Y for the left controller are pressed for 4 seconds to set the new coordinate system.

**Table 1: Overview of the ROS topic publisher and subscriber available from the Quest2ROS app.**

| Topic | Description | Sub/Pub | Msg type | Frequency |
|---|---|---|---|---|
| /q2r_left_hand_pose | Pose of the left hand controller given as a 3D vector and quaternions | Publisher | PoseStamped | 72.08 Hz |
| /q2r_left_hand_twist | Twist of the left hand given as a 3D vector and angular velocities around each axis | Publisher | Twist | 71.83 Hz |
| /q2r_left_hand_inputs | Inputs from the left hand controller | Publisher | Custom | 71.83 Hz |
| /q2r_right_hand_pose | Pose of the right hand controller given as a 3D vector and quaternions | Publisher | PoseStamped | 71.87 Hz |
| /q2r_right_hand_twist | Twist of the right hand given as a 3D vector and angular velocities around each axis | Publisher | Twist | 71.78 Hz |
| /q2r_right_hand_inputs | Inputs from the right hand controller | Publisher | Custom | 71.81 Hz |
| /q2r_left_hand_haptic_feedback | Haptic feedback control for the left hand | Subscriber | Custom | NA |
| /q2r_right_hand_haptic_feedback | Haptic feedback control for the right hand | Subscriber | Custom | NA |

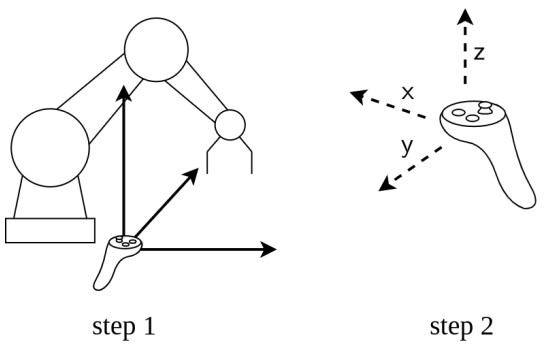

step 1                    step 2

**Figure 4: The figure shows the process of coordinate system alignment. First, either controller is aligned to the desired reference frame and the buttons A and B for the right controller or X and Y for the left controller is pressed (step 1). All telemetry data from the controllers are now given in the newly set coordinate system (step 2).**

## 2.3 Publisher and Subscriber

The position, rotation and velocities in and around each axes of the controllers are made available via ROS-topics. The coordinate axes of the controllers follow the right-hand rule, with x pointing forward, y to the left, and z pointing up , as shown in Figure 5.

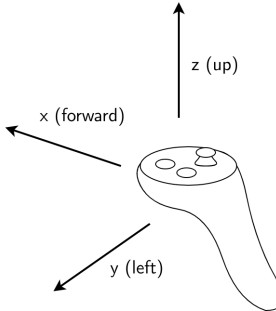

z (up)

x (forward)

y (left)

**Figure 5: The figure shows the positions and directions of the axes of the controllers. The axes follow the right-hand rule, with x pointing forward, y to the left, and z pointing up.**

The VR controller telemetry is given as separate pose and twist messages. The pose consists of two vectors

$$\vec{p} = \begin{bmatrix} x \\ y \\ z \end{bmatrix} \qquad \text{and} \qquad \vec{q} = \begin{bmatrix} q_w \\ q_x \\ q_y \\ q_z \end{bmatrix},$$

corresponding to the position and orientation, represented in quaternions, of the controller, respectively. The twist is given by the vectors

$$\vec{v} = \begin{bmatrix} v_x \\ v_y \\ v_z \end{bmatrix} \qquad \text{and} \qquad \vec{\omega} = \begin{bmatrix} \omega_x \\ \omega_y \\ \omega_z \end{bmatrix},$$

which represent the linear and angular velocity in and around each axis, respectively. Via ROS publishers, the user is able to access real-time pose, twist, and controller inputs, i.e., buttons' state, of left and right controllers. A subscriber is available for each VR controller to facilitate haptic feedback to the user by specifying the vibration of the controller. The full list of topics is reported in Table 1.

## 2.4 Brief instructions

To use Quest2ROS, the user needs to:

(1) Install and start the TCP-to-ROS endpoint on a host.
(2) Connect the VR headset to a Wi-Fi network from which the host is available.
(3) Install and start the Quest2ROS app on an Occulus Quest 2 or 3.
(4) Specify the TCP-to-ROS endpoint's IP address and port within the app and connect.
(5) Remove the headset from the head and place it where it has direct sight of the controllers by its cameras.
(6) (optional) Create a new reference frame within the room by aligning the right controller and pressing buttons A and B simultaneously for 4 seconds.

More details on the setup and use of Quest2ROS system can be found on the project website.

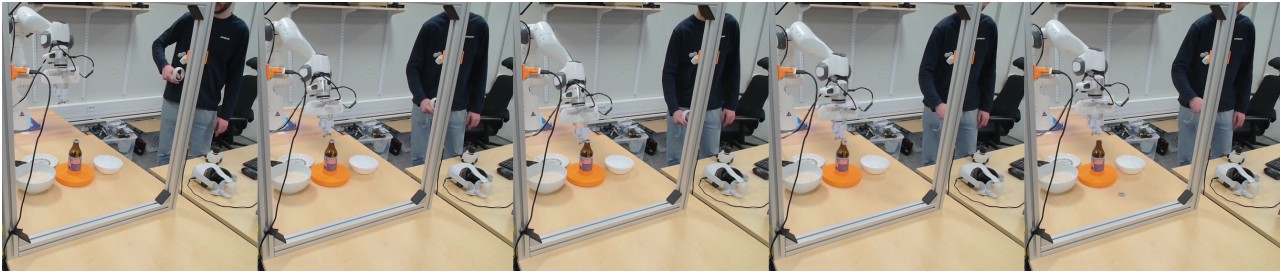

**Figure 6: Example usage of the Quest2ROS app on a bottle opening task using a Franka Panda Robot.**

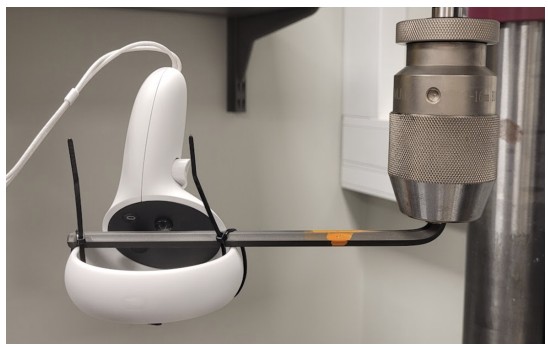

**Figure 7: The figure shows the setup used to fix the controller to follow a circular path used for accuracy evaluation.**

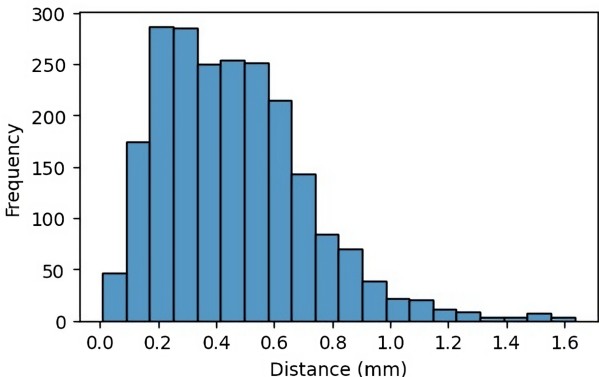

**Figure 8: The figure shows the distribution of distances of the recorded data points to the circle in the accuracy experiment.**

## 3 PERFORMANCE EVALUATION

To evaluate the performance of the Quest2ROS system, the accuracy, latency, and update frequency of the controller telemetry were measured. In this section, the methods and results of these experiments are presented.

**Accuracy:** The accuracy of the streamed controller data was evaluated by recording the measured positions while physically constraining a controller to follow a fixed circular path. By calculating how much the positional data deviates from the circle, we can assess how much the data fluctuates over time and space. The setup used to fix the controller to follow a circular path is shown in Figure 7. The controller was manually rotated four turns while the headset was placed at a distance of approximately 1 meter while facing the controller unobstructed resulting in 2180 datapoints. Since the position of the circle in the VR reference system is not known beforehand, the position of the circle was optimized to minimize the mean distance from each recorded data point.

The mean distance from the recorded data points to the optimized circle was 0.46 mm. A histogram showing the frequency of the recorded data points within different distance ranges is shown in Figure 8.

**Latency:** The latency of the streamed telemetry data was measured by recording the time from when a controller was physically moved till that the movement was registered in a subscriber ROS node. The time was measured using a high-speed camera with both the controller and the output of the ROS subscriber within its view. The high-speed camera used recorded 940 frames per second. The time was measured seven times using this method. The latency

measured in the eight tests were 106, 76, 89, 95, 79, 67, and 63 ms with a mean latency of 82 ms. The latency measurement videos are available on the project website.

**Frequency:** The update frequency of the telemetry data was measured with the built-in ROS command 'rostopic hz' on all publishers to the ROS node. The average measured update frequency was 71.86 Hz. The individual measurements for each topic are reported in table 1.

**Example usage:** We showcase the usability of the App by connecting it to a Panda Franka Robot which has a bottle opener on the end effector and is teleoperated to open a bottle as shown in Fig 6. We relay the twist message received on the topic *q2r_right_hand_twist* to a velocity cartesian controller which translates the received velocity to robot action as long as the left trigger button is held which is received via the *q2r_right_hand_inputs* topic. A full 30 min video of an operator performing the task is shown on the project website.

## 4 CONCLUSION

Quest2ROS lowers the entry barrier for researchers and engineers to perform intuitive control of high degree of freedom robotic systems. The Meta Quest 2 and 3 are readily available consumer-grade electronics, and ROS is a common framework of choice within the research field of robotics. Joining these technologies in a seamless solution results in a tool that has the potential to be highly useful within the community. Videos showing the framework being used on different robotic platforms are available on the project website.

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
