# OpenReview forum: "Quest2ROS: An App to Facilitate Teleoperating Robots"
_humanrobotinteraction.org/HRI/2024/Workshop/VAM-HRI — VAM-HRI 2024 Oral_

### Official Review · Reviewer_UDkj · 2024-02-20
**Accept**

**Rating:** 9
**Confidence:** 5

**Review:**

The paper presents a stand-alone application compatible with Oculus Quest 2 and 3, designed to enhance the teleoperation of robots via the Robot Operating System (ROS). The app facilitates the publication of hand-held controller positions, velocities, button presses, and haptic feedback. It offers a simple method for aligning the coordinate frame of the controller with any robot in the real world, boasting an average tracking accuracy of 0.46 mm, a mean latency of 82 ms between the Quest 2 and a ROS node, and an update frequency of 71.96 Hz for relevant ROS topics.

## Strengths:
- **Innovative Integration:** Combines consumer-grade VR hardware with ROS to make robot teleoperation more intuitive and accessible.
- **High Precision and Responsiveness:** Demonstrates high tracking accuracy and reasonable latency, which are critical for delicate teleoperation tasks.
- **Flexibility and Robot Agnosticism:** The app's ability to easily align coordinate frames and its compatibility with different robots enhance its utility across various teleoperation scenarios.

## Weaknesses:
- **Limited Evaluation:** While the paper presents performance metrics, a broader evaluation including user studies could better assess usability and practical efficiency.
- **Hardware Dependency:** The reliance on specific VR hardware (Oculus Quest 2 and 3) might limit accessibility for users with different devices.

## Recommendations for Improvement:
- **Extend Hardware Support:** Develop versions for other VR/AR systems to broaden the application's accessibility and utility.
- **Comprehensive User Studies:** Conduct extensive user studies to evaluate the app's usability, effectiveness, and impact on teleoperation tasks across diverse user groups.
- **Enhance Documentation and Support:** Provide detailed documentation and tutorials to facilitate the adoption of Quest2ROS by researchers and practitioners in robotics.

In summary, I think this paper is a great fit for VAM-HRI, and I recommend acceptance.

---

### Official Review · Reviewer_m6gB · 2024-02-26
**Accept**

**Rating:** 7
**Confidence:** 5

**Review:**

This paper presents Quest2ROS, a publicly available stand-alone app for the Quest 2 and Quest 3 virtual reality headsets that enables position tracking-based teleoperation of manipulator robots, through the movement of handheld controllers. The functionality and architecture of the Quest2ROS system is described, and an analysis of the system's tracking accuracy is conducted.

Strengths:
-  This paper introduces a publicly available instantiation of a component (live teleoperation) useful for a wide variety of VAM architectures, providing future researchers with the option to reduce their engineering efforts through the use of this off-the-shelf system.
- The position tracking evaluation is well-designed, and allows the accuracy of the system to be precisely measured under ideal conditions. These measurements are very useful for developers moving forward.

Weaknesses:
- The paper mentions that the headset does not need to be worn by the operator. This does seem helpful for certain task setups, where the operator is able to directly observe the robot performing its actions in real time from a close distance. However, the paper doesn't describe what (if anything) is visualized within the VR headset if it is worn while the teleoperation is underway. Does the system work effectively when the operator is not in the same room as the robot?
- The paper does not include any quantitative or qualitative evaluations about the human-factors side of the teleoperation (i.e., how easy is it to use), especially as compared with other teleoperation systems. This is good content to include in future work.
- Needing to place the headset so that it has direct line of sight to the controllers at all times seems like it might introduce some issues during actual use - this should also be explored in future human-subjects studies.

Minor notes:
- The paper could include some more details as to what the purpose of the reference frame is: does this match the current 6DOF pose of the controller to the current 6DOF pose of the end effector of the teleoperated robot? Of some other reference pose? What does resetting the reference frame do in terms of affecting the movement of the robot?
- A couple minor typos in Section 1: "pupped" should probably be "puppet," and the sentence "...the app is available for the Oculus Quest 2 and 3 and allows to use the VR controller to teleoperate the robot..." is worded strangely - "allows to use" should probably be "allows the use."

Summary:
This paper is clearly relevant to the VAM-HRI workshop and makes a easily recognizable contribution to the field at large. I recommend it for acceptance and presentation at this year's workshop.

---

### Decision · Program_Chairs · 2024-02-26

Accept (Oral)